
# The catastrophe of the Niedów dam – the dam break causes, development and consequences

Stanisław Kostecki[1,*] and Robert Banasiak[2,*]

[1]Faculty of Civil Engineering, Wrocław University of Science and Technology
[2]Institute of Metheorology and Hydrology and Water Management – National Research Institute

**Correspondence:** Robert Banasiak (Robert.Banasiak@imgw.pl)

**Abstract.** Due to extreme rainfalls in 2010 in the Lusatian Neisse river catchment, a flood event with a return period of over 100 years took place leading to the failure of the Niedów dam. The earth type dam was washed away with a result of a rapid release of nearly 8.5 million $m^3$ of water volume and flooding of the downstream area with substantial material losses. The paper analyses the conditions and causes of the dam's failure, while special attention is given to the mechanism and dynamics of the compound breaching process. Several empirical formulas for the dam breach prediction are tested with regards to their usefulness in assessing breach dimensions and outflow peak discharge. The paper also describes a numerical approach to simulate the flood propagation downstream the dam with the use of a two-dimensional hydrodynamic model. Considering the specific local conditions and available data set, an iterative, reversed solution of the unsteady state problem is made. This approach enabled to deliver realistic flood propagation estimates, verification of the dam breach outflow, and several important answers on the dam failure consequences.

## 1 Introduction

Worldwide, there is increasing number of dams for water storage and supply to meet the growing demand from towns, agriculture, industry or power generation. An important role is also the flood risk mitigation. Apart from the substantial benefits to the society provided by dams, there is also an inherent and growing risk of dam failure. This results in flooding with potential serious material damage and loss of life. In the twenty-first century, around 200 notable dam and reservoir failures occurred worldwide (Cannata and Marzocchi, 2012). A spectacular example is the case of the world's largest dam failure of the Banqiao Dam, Shimantan Dam in Henan Province in China in 1975 causing about 26,000 human deaths (Peng and Zhang, 2012). The causes for the dam failure may be the faults during the design and construction stages, ageing of the structures and climate change resulting in altering of meteorological and hydrological patterns. Many dams were already built at the beginning of the XX century, and are now beyond the end of their projected lifespan (Grant, 2001; Hansen et al., 2020). Climate change





causes the occurrence of local extreme hydrologic events that may exceed the physical capabilities of the engineered spillways of aging dams (Hansen et al., 2020; Ho et al., 2017).

Most of existing dams can be classified as large dams. Two thirds of them are earth fill dams. The earth dams are first choice
for construction for relatively low construction costs, possible placement on non-rock basements, and for being capable to resists uneven settlement of the dam body. On the other hand, this kind of dam is prone to damages due to overtopping or piping through the dam body or the ground underneath. Thus, the failures of the dam embankments cover almost 80% of the accidents and the remaining 20% are referred to concrete dams. Zhang et al. (2009) and Cleary et al. (2015) reported that around 88% of the dam breaks occurring from 1989 to 2006 in China were of earth and earth-rock dams. The primary cause for
the embankment dam failure is overtopping, which constitutes 50% of embankment dam failures when the piping constitutes about 34% (ICOLD, 1995; Zhang and Tan, 2014).

The key data for the forecast of dam failure consequences in terms of downstream inundation time, depth and extend of possible damage, is the outflow hydrograph. The hydrograph shape, volume and peak outflow depend on the evolution of the dam break, height of the dam, reservoir storage volume. Typically, basic geometry parameters of the dam breach are the
breach size, breaching geometry and dynamics, which is typically very high, and the discharge coefficient for the breach. This coefficient further depends upon the breach width and length (transverse and longitudinal dimensions), sort of material and reinforcement of the dam, height of the water level respective the breach crest and its irregularity.

The development of the dam breach can be assessed relatively easily and properly only for the simple cases, i.e. with uniform dam bodies. However, the earth dam structures are typically more complex, consisting of a number of different material layers,
equipped with sealing cores, drainage facilities and wave protectors, even with paved roads on the top of dam. Hence, in most practical attempts, the prediction of the dam breach is much more complicated, and a number of methodological approximations and simplifications were adopted.

An early and most common in use method is the multiple linear or nonlinear regression analysis, utilizing data collected from actual dam failures, without considering the complex physical laws governing the natural process. Upon this developed
regression formulas link the essential data regarding the dam construction, the mode of failure, average width of the embankment between toes of the downstream and upstream slopes, water volume above the breach bottom at the time of failure, height of water above the breach bottom at the time of failure, the breach average height and width, breach side slope ratio of a trapezoidal approximation of the breach, and the formation time of the breach. The most commonly estimated parameters of breach formation include the reservoir water-surface elevation at which breach formation begins or the critical overtopping
depth, breach formation time, the breach height and width, and average side slope ratio (Wahl, 2004; Froehlich, 2008; Ashraf et al., 2018). Zhang et al. (2009) applied multi-parameter nonlinear regression analysis and found out that dam erodibility is the most important factor, influencing the other breaching parameters and that the equations for predicting the breach geometric parameters appear to be more accurate, than those for predicting the peak outflow rate and the failure time. Another approach relies on the peak outflow prediction as a function of various dam and/or reservoir parameters, instead of breach geometry and
time (Bureau of Reclamation, 1982, 1988; MacDonald and Langridge-Monopolis, 1984; Froehlich, 1995a; Macchione, 2008).



Physically based models involve formulas for the erosion of the dam forming material or coupled equations for sediment transport and shallow water equations in a one dimensional (1D) form (e.g., Tingsanchali and Chinnarasri, 2001) or two-dimensional form (2D). E.g. Zhong et al. (2016) utilized and compared three simplified physically based models: NWS Breach, HR Breach and DLBreach and found out that the DLBreach model was found to be the most accurate, because it uses total-load nonequilibrium transport equation and headcut migration rate formula for non-cohesive and cohesive embankment breaching, respectively. Wang and Bowles (2006a) developed an erosion and force and moment equilibrium based three-dimensional dam breach model for the non-cohesive earthen dam overtopping breach and validated it using a field test (Wang and Bowles, 2006b).

Use of the physically based solutions can provide better understanding of the mechanism of a breach. However, they are complex and require several assumptions, e.g. related to the breach shape, which is assumed to vary from triangular to trapezoidal as the breach progresses (Wahl, 1998). Also an irregular shape of the breach was proposed (Tsakiris and Spiliotis, 2013). Fundamental for the determination of the outflow hydrograph is the breaching time, which depends on the dam material properties, mode of breach (overflow or piping), which defines the evolution of the breach in a streamwise direction and/or the conditions for stability of the slopes of the breach. Assumptions made simplifying the mechanism of the breach can result in incorrect simulation of the observed case studies (Elmazoghi, 2013).

To better understand the mechanism of the breach of earth dam, numerous experimental studies were carried out in the laboratory. (Chinnarasri et al., 2004) performed numerous tests and derived an equation for estimating the breach deformation time. Orendorff et al. (2013) quantified the effect of initial breach conditions on breach parameters. Ashraf et al. (2018) used the result of laboratory tests to verify a new set of equations of breach parameters obtained on the basis of non-linear regression analysis of actual failures of embankment dams. Nourani et al. (2012) used laboratory tests and historical data to develop two ANN models to simulate outflow hydrograph from the breach. Liu (2019) applied the large-scale particle image velocimetry to measuring surface velocities in earth dam-break experiments.

All methods presented did not consider the earth dam with the upper slope protected against wind waves, ice cover and seepage through the embankment. Only Zhang et al. (2009) and Xu and Zhang (2009) distinguished concrete faced kind of earth dam in their empirical prediction models. In Poland, for example, every earthen dam is concrete or asphalt faced to protect against the harmful effects of ice and water waves. Typically, the water side slope, is protected by concrete slabs with expansion joints bitumen sealing. On top of the dam there is so called technical road, or road connected to the open traffic. Such a construction is likely to significantly affect the breach time and breach geometry, hence it affects the outflow hydrogram and peak.

The evaluation of the risk of failure and its consequences relies on testing of a number of the catastrophe scenarios in order to analyze and assess the consequences of the potential flood. The base of such analysis are hydrologic simulations, numerical modelling of breaching processes and flood plain flows as well as inundation maps and GIS systems (Altinakar, 2008; Cleary et al., 2012, 2015; Cannata and Marzocchi, 2012; Álvarez, 2017; Zhong, 2011). 1D models can predict the flood propagation in channels with reasonable accuracy and good efficiency, but a 2D or hybrid1D/2D approach should be used in wide floodplains and complex terrain regions with elevated roads, secondary dikes, levees, buildings, and other obstacles





((Vanderkimpen and Peeters, 2008)). Although 1D models are most used in flood routing, the 2D models gain applications due to significant computational power advances in recent years (Saberi et al., 2013; Yakti et al., 2018). Also hybrid models are used to derive 1D based breach outflow hydrographs whereas 2D model is used for flood plain modelling and generation of inundation maps downstream the dam (Shah et al., 2019).

The current work presents a case study of a catastrophic failure of the Niedów dam, in Poland, in 2010. A detailed picture of the dam break evolution is based on witnessed, stored or restored data. The geographic, meteo- and hydrological conditions leading to this event are presented. In particular, the description of the failure mechanism of the reinforced earth dam is highlighted. The work further makes use of different methods and selected formulas, for assessment of the dam breach along with determination of the peak outflow. Finally, a two-dimensional (2D) hydrodynamic model is applied to validate the formula

outputs and to clarify the flood wave propagation, in particular, quantifying the effect of the Niedów dam failure on the flooding downstream area.

## 2   Study Site

### 2.1   Cartographic Information

The Niedów dam on the Witka river (in km 2.2) is located in the south-west Poland, near the Polish-Czech and Polish-German

borders. It was constructed in 1962 to supply water to the Turów coal power station, for cooling purposes and for drinking water supply to nearby settlements, including the town of Bogatynia. Basically, the reservoir function was not to mitigate the flood hazard. The storage capacity of the reservoir before the failure – at normal water level of 210 m a.s.l. – was 5.6 million $m^3$ and the water surface area was that of 183 hectares. The reservoir catchment area is 321 $km^2$ in the (sub)mountainous region of the Izerskie mountains, with significant head water slopes. Most of the catchment is located on the Czech Republic

territory as shown in Fig. 1. The geological structure of the river bed are granite and gneiss formations under a layer of sands, gravels and clay, locally. Such formations are favorable to high run-off.

### 2.2   Meteorological and Hydrological Conditions

In the period of 6 to 8 August 2010, the upper catchment of the Lusatian Neisse (in Polish – Nysa Łużycka) was subjected to exceptionally high amounts of rainfall. In the Witka catchment, a tributary of the Lusatian Neisse, a rainfall sum of two

days reached values in the range of 150-250 mm, whereas the daily sum of the 7th August reached values of 128.5 and 179 mm in the metheorological stations Mnisek and Heinice, respectively, see Fig. 2. The most intensive rainfall occurred in the morning between 8 and 9 a.m., with an hourly rainfall of 15-35 mm, locally reaching even 58 mm (Heinice station). This rainfall statistic corresponds to one fourth of the yearly rainfall in this mountainous region. Moreover, the hydrological situation deteriorated due to the precedent wet period in the second half of July with precipitations above the norm, which led

to the saturation of the ground and acceleration of the run-off later on. The consequence of such meteorological situation was the appearance of catastrophic floods on several rivers, including the flash floods on the Witka river, the Miedzianka river and


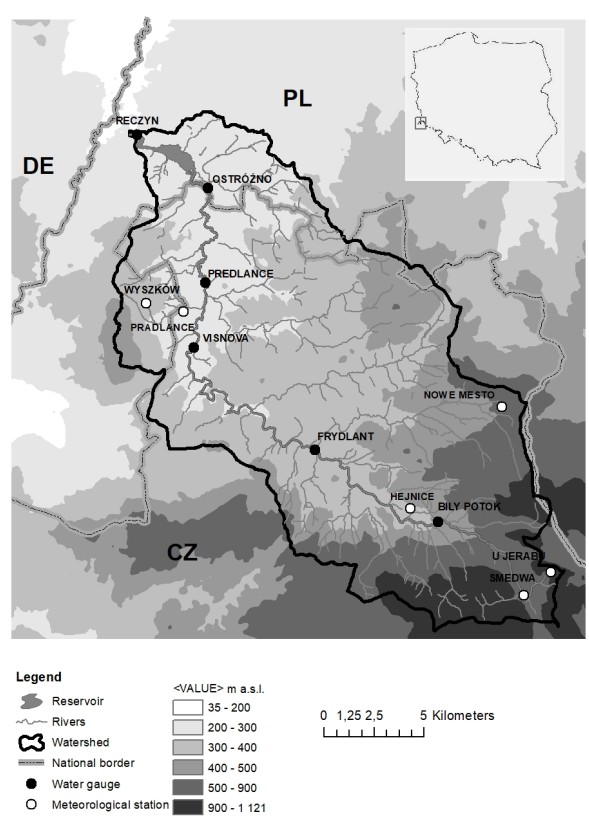

**Figure 1.** The catchment of the Witka river

on the major Lusatian Neisse river (IMGW, 2011). In most hydrological gauge stations, observed water levels significantly exceeded historical maxima. Remarkably, a number of gauge meters was destroyed during the floodwater passage, making it more difficult to subsequently assess the quantitative data of the flood. The return period of the flood is estimated to be within
100-200 years.

The flow on the Witka river - its name on the Czech territory is the Smeda river - is monitored at four gauge stations. On the Polish section from km 0.0 (river mouth) to km 8.0, there are two stations: Ostróżno (km 7.98), upstream the reservoir, and Ręczyn (km 1.8), downstream the reservoir (Fig. 1). On the 7 August at the gauge Ostróżno the highest water level occurred at 16:40. The gauge station Ręczyn recorded the water levels until 15:20, as it was destroyed due to high release of water
from the reservoir, yet before the dam break. During the 45-year period of continuous flow monitoring at Ostróżno gauging station, the flood discharges were less than 70 $\text{m}^3\text{s}^{-1}$, which still is within the limit of bankful flow. There was just a single higher flow in August 2001, equal to 171 $\text{m}^3\text{s}^{-1}$. That event also featured a rapid ascent and descent of the wave, typical for the flash flood. On the critical 7 August the estimated flood rate was 615 $\text{m}^3\text{s}^{-1}$, but this estimation is still burdened with





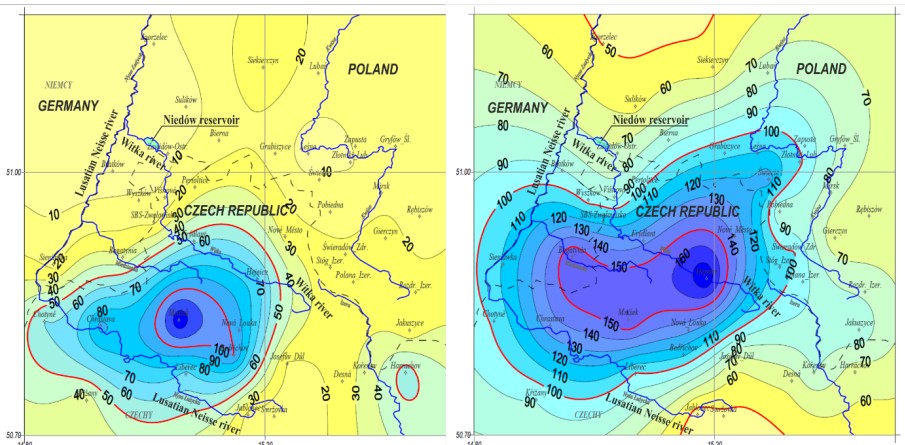

**Figure 2.** Rainfall values for 6.08.2010 (left) and 7.08.2010 (right) (in mm)

significant uncertainty and was a subject to discussion. A direct reliable estimation of the peak flow rate was not possible, as
the water level substantially exceeded the measuring range. In addition, the topography makes it more difficult due to locally
wide floodplain. Between the Ostróżno cross-section and the reservoir, there is an increase of the catchment area from 268 to
331 $\mathrm{km}^2$, including the Koci Potok stream, which also severely flooded and delivered a significant direct inflow to the reservoir.
This stream was not monitored, but based on the field survey after the flood, the peak flow rate was estimated to ca. 70 $\mathrm{m}^3\mathrm{s}^{-1}$.

## 2.3  Field observations

A field survey was carried out past the flood to collect data on the flood wave passage along the Lusatian Neisse river and
its major tributaries (IMGW et al., 2010). A number of eye witnesses of the flood were interviewed, including several local
authority representatives. Water marks were searched and fixed for further geodesy recording. Maximum water elevation marks
were recorded in over 50 locations, including upstream and downstream the Niedów dam. It is to be pointed out that post-factum
determination of the exact maximum water elevation is not a simple task. In some cases, clear water lines were found on walls
in a form of sediment and residues marks or washed out dirt, but in several locations high water marks were approximate,
based on residues found on threes, bridge piers or decks. Therefore, the error of maximum water elevation may vary from a
few millimeters to ca. 0.3 m. Apart of that, the limits of flooded area were also explored and brought on the map, consecutively
digitalized. This later served also as a reference for the verification of the flood hazard maps elaborated in the context of the
EU Flood Directive.

An important information is the timing of the flood on the Lusatian Neisse river at the section of the Witka river inflow as
no gauge is located nearby. Based on inhabitants reporting it has been found that culmination of the flood occurred between 2
and 3 a.m. on the 8 August 2010, hence significantly later than the peak outflow from the Niedów dam. This information helps
explaining the effect of the coincidence of the flood waves from the two rivers and to find proper upper boundary conditions
for the hydrodynamic model.





It is also worth mentioning here about a good communication and cooperation between Polish and German (Land of Saxony) water services and authorities to put under discussion and gather the needed information, and reaching an agreement on selected important data. In this respect, a bilateral experts group was set up to deliver a common set of data to be used in hydraulic modelling. As a result, a common 2D modelling platform is being developed for consistent flood hazard mapping for this transboundary territory (ICPO, 2019).

**2.4    Description of the Niedów Dam**

The Niedów dam consisted of three major sections: the central section with a concrete water release structure equipped with movable gates, bottom outlet and a hydropower plant, with the total length of 47.05 m. Two other sections were earth embankments on the left and right side with a length of 126 and 94 m, respectively, see Fig. 3.

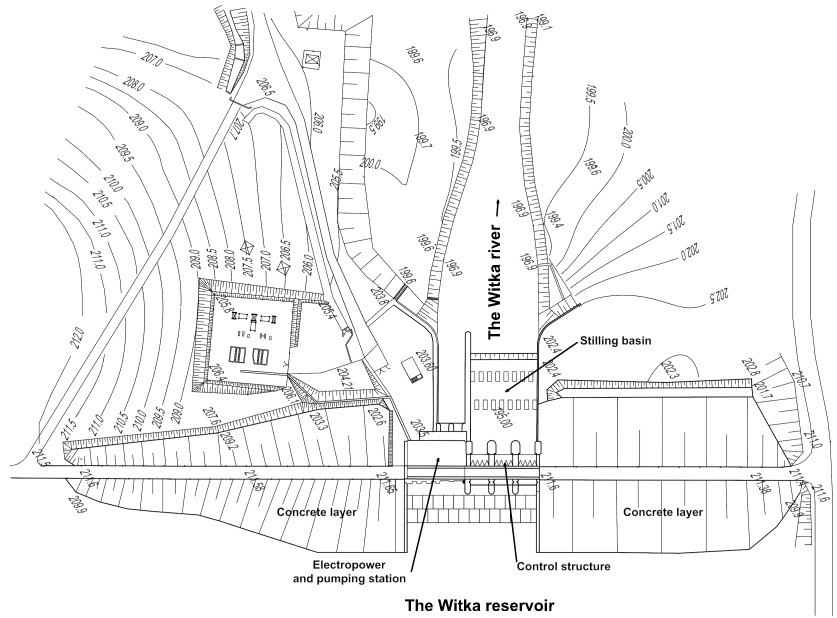

**Figure 3.** Plan view of the Niedów Dam

    Three tainter steel gates, with a width of 6.7 m and a height of 6.6 m each, controlled the water outflow from the reservoir,
see Fig. 4. The pillars contained bottom outlet openings with size of 2x1 $m^2$, equipped with vertically moving flat closures.
    The structure and geometry of the earth dam are depicted on Fig. 5. The maximum height of the embankment in respect to the base ground level was 11.6 m. The body of the dam was well compacted sand. The slope upstream was of a ratio 1:3, while the slope downstream of 1:2.5. The total volume of earth dam was ca. 61,000 $m^3$. Because the sand had a high filtration rate of 2.8×10-3 $ms^{-1}$, the upstream slope was shielded with a double layer of concrete slabs with dimensions of 1.5×1.5×0.1 m,
sealed with a bituminous material. This shield from upstream was supported by a vertical reinforced concrete filtration screen, reaching down the basement rock. The downstream slope was covered with grass on humus. In the lower part of the slope a





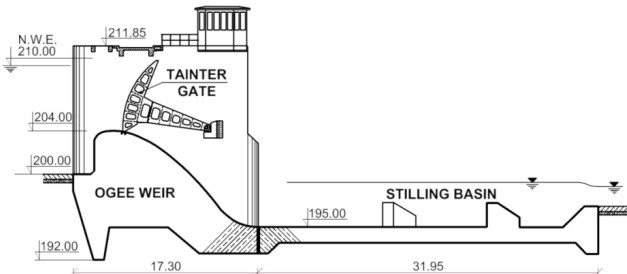

**Figure 4.** Ogee weir cross-section

drainage of mixed gravel and stone was provided. The dam crest was 5 m wide and served as a road made of concrete slabs with asphalt. The power plant and water outlet sections were connected with the dam by means of abutments.

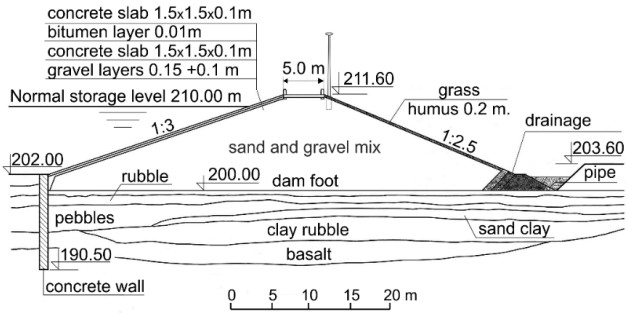

**Figure 5.** Cross-section of the earth dam

## 2.5 Dam Wash out Mechanism and Breach Characteristics

The process of destruction of the Niedów dam followed a specific pattern as the overflow and washing out was different for the left and right dam side, both, in terms of timing, and dynamics. An S-shape plan of the reservoir just upstream the dam (Fig. 1) caused a low angle of inflow direction to the dam axis. As a result, the water level at the left side was higher several centimetres than on the other side, leading to uneven overflow. The overflow started at 17:00 over the left dam near the bank because of a slight crest inclination towards it. The water passing over the crest caused the erosion first around the lighting foundations
(additional turbulences) and progressive washing out of the downstream, grass grown slope. This action took approximately half an hour resulting in a damage of the dam crest, by disintegrating it part by part, since the road concrete slabs lost support of the eroded sand. Further, the dam breach moved regressively upstream causing upstream slope to disintegrate from the top. Remarkably, the concrete slabs when losing support broke in series like a chocolate, and were swept away by intensified flow. This phase was slowed down for a while as two concrete slabs resisted longer, hydraulically acting as a sharp crested weir.
Next, another important moment took place. As the support of the earth embankment vanished, the left training wall flanking



the central concrete dam collapsed due to upstream water pressure. This resulted in a further rapid outbreak. This phase was relatively short but intense, with a result of torrential flood wave downstream documented by Fig. 6. After next 80 min. the earth dam was almost completely swept away (Fig. 7).

The overtopping of the right dam begun approximately 15 min. after the left one. The breaching in this case developed in a similar but in less dynamic fashion as above. The wash-out started at the central part of the right dam, evolving towards the right bank, conforming with the water inflow direction. As a results of the fall of left abutment and a rapid lowering of the water level in the reservoir, the right bank washing out decelerated. In addition, the concrete slabs resisted to fail and worked as a weir for some 20 min. It is difficult to explain the origin of it. Possibly, the slabs jammed or the concrete debris temporally hindered the erosion progress. Fig. 6 again portrays this situation of hindered breaching of the right dam side. Finally, the

concrete slabs got down. Nevertheless, the outflow here was not that intense any more, since the upstream water level had already substantially decreased. The washing out of the right dam lasted for about 130 min, causing a devastation of 62 per cent of its length, reaching the base level of 200.00 m a.s.l. The final width of this breach was 58 m as illustrated by Fig. 8. A part of the dam adjacent to the control structure remained. Table 1 has the crucial moments of the breaching development, established via blending of observations, records, and interviews.

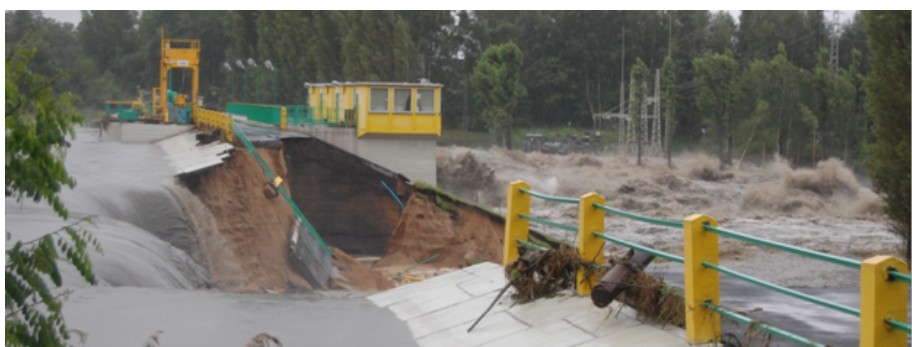

**Figure 6.** View from the right bank. Behind the concrete structure an immense outflow is visible after the abutment had collapsed. Water level in the reservoir ca. 210.60 m a.s.l.

The breach parameters, i.e. the eroded earth volume $V_{er}$, the mean breach width $B_{avg}$, breaching time $T_f$, are collected in Tab. 2. respective both to the left and right embankment. For use of empirical formulas, the breach depth, erosion time and reservoir volume above the breach is calculated as well.

## 3    Methods

### 3.1    Empirical formulas for Breach Calculation

The breaching development prediction is essential for making the estimates of flood propagation and flood hazard. Therefore, with own data a number of empirical and regression formulas are put to the test to calculate breach characteristics, including the





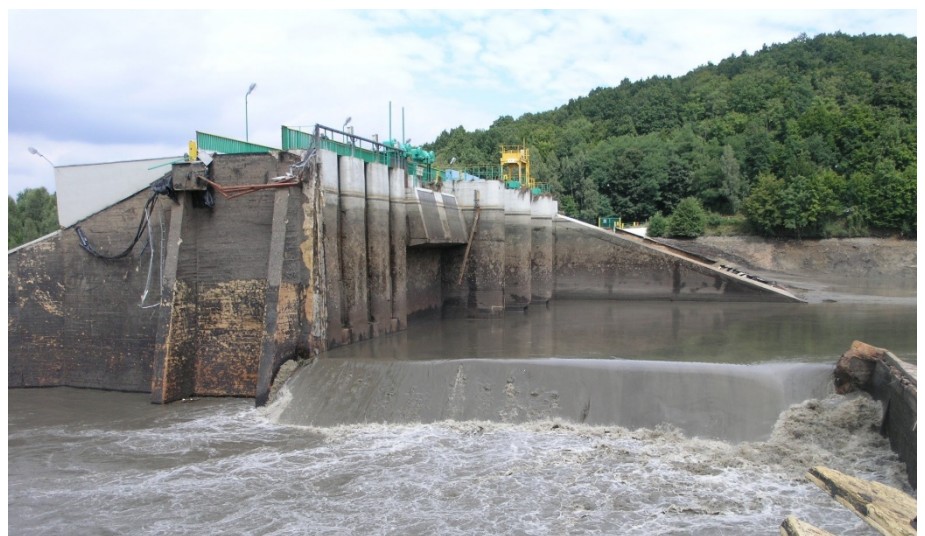

**Figure 7.** Broken left retaining wall of the control structure

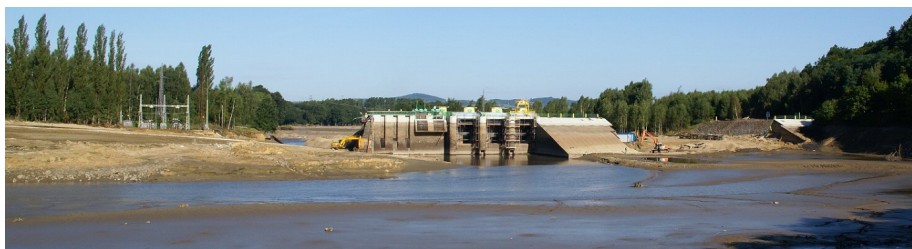

**Figure 8.** The final breach of the dam – an upstream view

breach size, time and peak outflow. The ultimate selection of the formulas took into consideration the following aspects: first, the underlying data set relates to a significant number of case studies and different types of dams, second, a promising value of the correlation coefficient. As a result, six approaches developed by different authors are adopted herein. The formulas used for

particular breach characteristics, as to refer to the Niedów dam observation, are collected in Table 3-6. The formulas differ in terms of their form and parameters. An important aspect is that most of the formulas do not properly refer to the concrete-faced dams, although several dams of similar kind were present in the underlying data. An exception here are formulas by Xu and Zhang (2009), Zhang et al. (2009), introducing a correction factor for the dam reinforcement, denoted to as $B_3$.

The quantities in these equations and their units follow the description provided in Table 2. Appearing in the equations of

Xu and Zhang (2009) $H_r$ is a reference dam height set as 15 m and $T_r$ is a reference failure time set as 1.0 h, $B_1$ means maximum depth behind dam at the breach inception. The coefficients $B_3$, $B_4$ and $B_5$ are related to dam type (classified as dam with corewall, concrete-faced dam and homogeneous or zoned-fill dam), dam erodibility (high, middle and low), failure



**Table 1.** The failure development of the Niedów dam on 7 August 2010

| Time | Development |
|---|---|
| 15.00 | Outflow from the reservoir - 86 $\mathrm{m^3s^{-1}}$, WL - 210,02 m a.s.l., gates I, II, III open 60 cm |
| 15.36 | Water inflow into the power plant and in the control room, the crew evacuation and manual opening of the gates |
| 15.50 | Outflow 140 $\mathrm{m^3s^{-1}}$, WL 210,21 m a.s.l., gates I, II i III open 150 cm |
| 16.10 | A rapid rise of the water level in the reservoir |
| 16:40 | The maximum water level at Ostróżno gauge station |
| 17.00 | Beginning of the flow over the left dam, the crew evacuated, gates I and III open 250 cm, II open 200 cm, outflow 352 $\mathrm{m^3s^{-1}}$ |
| 17.15 | Beginning of the flow over the right dam |
| 17.42 | Water level reaches a maximum of 212.05 m a.s.l. Washing-out of the lee sides of the dam, destruction of the road on top of the dam – 40 m on the left side and 30 m on the right side |
| 18.10 | Water level at 211.60 m a.s.l. The breaching likely reached the dam floor. The collapse of the headwall of the left dam resulting in immense outflow through the breach |
| 18.47 | The breach of the left dam is finished. The dam is washed out on length of 106 m |
| 18.56 | Water level - 209.00 ma.s.l. The right dam continues breaching |
| 19.25 | The breach of the right dam is complete. The dam is washed out on the width of 58 m. The reservoir releases remaining water |
| 21.00 | The reservoir got empty |

type (overtopping and piping). In our case, to establish these coefficients, the following assumptions were made: overtopping, concrete-faced dam and highly erosive material of embankments. This led to $B_3 = 0.399$, $B_4 = -1.303$, $B_5 = -2.458$.

Since the breach for the left and right embankment developed differently, it was necessary to calculate or estimate the input parameters, i.e. the reservoir storage to the crest level and water volume above the breach bottom at the time of failure regarding both embankments. The in time varying outflows of the control structure (all three tainter gates) and the two breaches were calculated using a well-known hydraulic formulas. To calculate the outflow thtough the tainter gates, the discharge coefficients used are in accordance with those obtained from physical laboratory model testing for Niedów dam (Herrera-Granados and

Kostecki, 2016). For the dam breaches, first a broad crested weir formula was applied until the road on the dam crest collapses, consecutively, a formula for the sharp crest weir inclined at an angle of 18 degrees (as the inclination of the concrete slabs), finally, a formula for no crest weir was used for the situation of breaching to the ground floor. The growing width of the breach was interpolated based on the photographs and films made during the catastrophe. The breach depth in specific moments, being the most difficult parameter to reconstruct, was calculated making use of the discharge-water level relationship for a section

closely downstream the dam delivered by the hydrodynamic 2D model. Based on the outflows volumes of the left and right breach, values of $V$ and $V_w$ were estimated. The following results are presented in Section 4.




**Table 2.** Dam failure characteristics

| HYDRAULIC CHARACTERISTICS | | | | |
|---|---|---|---|---|
| Reservoir storage to the crest level | Surface area to the crest level | Volume stored related to $H_w$ | Depth behind dam at breach inception | Peak discharge |
| $V$ | $A$ | $V_w$ | $H_w$ | $Q_p$ |
| million m$^3$ | m$^2$ | m$^3$ | m | m$^3$s$^{-1}$ |
| 8.310 | 1,900,000 | 8,541,000 | 11,70 | 1356 |

| EMBANKMENT DIMENSIONS | | | | |
|---|---|---|---|---|
| | | | LEFT DAM | RIGHT DAM |
| Max height[a] | $H_d$ | m | 11.6 | 11.6 |
| Crest width | $W_c$ | m | 5.0 | 5.0 |
| Bottom width | $W_b$ | m | 68.8 | 68.8 |
| Average width | $W_{avg}$ | m | 43.5 | 57.25 |
| Upstream slope | $Z$:1 | - | 0.333 | 0.333 |
| Downstream slope | $Z$:1 | - | 0.4 | 0.4 |

| BREACH CHARACTERISTICS | | | | |
|---|---|---|---|---|
| | | | LEFT DAM | RIGHT DAM |
| Depth above breach (max) | $H_b$ | m | 11.6 | 11.6 |
| Average depth[b] | $H_{bavg}$ | m | 6.0 | 9.7 |
| Top width | $B_t$ | m | 106.0 | 58.0 |
| Bottom width | $B_b$ | m | 21.8 | 58.0 |
| Average width | $B_{avg}$ | m | 64.0 | 58.0 |
| Eroded volume | $V_{er}$ | m$^3$ | 25,130 | 20,860 |
| Breach formation time[c] | $T_f$ | h | 1.78 | 2.16 |
| Empty time[d] | $T_e$ | h | 3.00 | 2.75 |

[a] Difference between natural ground lowest point behind dam and dam crest level. [b] Average depth along the left and right dam axis respectively. [c] Breach formation times provided by (Froehlich, 2008). Considered to be "length of time needed for the final trapezoidal breach to form, which takes place after the breach initiation phase". [d] Times from overflow to empty reservoir.

## 3.2 2D Hydraulic Model for Flood Routing

A hydrodynamic 2D model was extended to determine temporal and spatial variations of the flood wave along the Witka river and the Lusatian Neisse river downstream the Niedów dam. The 2D model domain stretches from a section located several hundred meters upstream the Witka river mouth to the Zgorzelec city, most downstream. The Witka river is modelled along with a mill channel running through the Radomierzyce village. The tributaries end sections of the Pliessnitz river and the Czerwona





**Table 3.** Formulas to calculate the breach average width $B_{avg}$

| Reference | Equation |
|-----------|----------|
| MacDonald and Langridge-Monopolis (1984) | $B_{avg} = V_{er}/(H_b W_{avg})$ |
| Bureau of Reclamation (1988) | $B_{avg} = 3H_w$ |
| Froehlich (2008) | $B_{avg} = 0.27k_0 V_w^{0.32} H_b^{0.04}, \quad k_0 = 1.3$ (overtopping) |
| Xu and Zhang (2009) | $B_{avg}/H_b = 0.787(H_d/H_r)^{0.133}\left(V_w^{(1/3)}/H_w\right)^{0.652} e^{B_3}$ |
| Soliman (2015) | $B_{avg} = 48.644 V^{0.275} W_{avg}^{-0.086}$ |
| Ashraf et al. (2018) | $B_{avg} = 13.197 H_d^{0.4757} V^{0.1785}$ |

**Table 4.** Formulas to calculate the breach average depth $H_b$ and eroded volume $V_{er}$

| Reference | Equation |
|-----------|----------|
| MacDonald and Langridge-Monopolis (1984) | $V_{er} = 0.0261(H_w V_w)^{0.769}$ |
| Xu and Zhang (2009) | $H_b/H_d = 0.453 - 0.025(H_d/H_r) + B_1$ |
| Soliman (2015) | $H_b = 1.093 H_d^{0.894} V^{0.027}$ |
| Ashraf et al. (2018) | $H_b = 0.9067 H_d^{1.0118} V^{0.013}$ |

Woda river are included as well. The major task was to recognize and verify important local unknown or uncertain hydrological and hydraulic characteristics of the flood, including the basic outflow hydrogram from the reservoir, and its consequences. An important aspect of the flood routing was also to properly restore the overflow through the left embankments to the Berdzorfer

Lake, artificial post mining lake, which received a substantial amount of water due to the flow over the left embankment.

The hydrological system under concern is schematized on Fig. 9, a part of which is implemented into the 2D model. In essence, the purpose of this flood routing was balancing the inflow, outflow, and storage in the domain making use of the continuity and mass conservation principle. The unsteady state conditions were defined as follows:

$$Q_{NL,in}(t) + Q_{ND}(t) + Q_P(t) + Q_{CW}(t) - Q_B(t) = Q_{NL,Z}(t) \tag{1}$$

where: $Q_{NL,in}(t)$ – discharge hydrograph for the Lusatian Neisse as the upper boundary condition – preliminarily interpolated from the two gauge stations upstream and downstream (Sieniawka and Zgorzelec, the Rosenthal gauge was destroyed), but uncertain, to be verified; $Q_{ND}(t)$ – the Niedów outflow hydrogram – unknown; $Q_P(t)$, $Q_{CW}(t)$ – the flow rate hydrographs for the tributaries the Pliessnitz river and the Czerwona Woda river – known; $Q_B(t)$ – inflow to the Berzdorf lake – only the total volume of inflow is known to be in a range from 3.5 to 4 mln m3; $Q_{NL,Z}(t)$ – the discharge hydrograph for the Zgorzelec

gauging station; the discharge is calculated based on stage hydrograph and the rating curve, nevertheless, the high discharge values must be considered with a margin of error due to extrapolation.

Importantly, a hydrometric discharge measurement took place at Zgorzelec cross–section during the maximum water level. Under difficult measuring conditions a discharge rate of 1040 m$^3$s$^{-1}$ was obtained. This is somewhat more than the value



**Table 5.** Formulas to calculate the breaching time $T_f$

| Reference | Equation |
|---|---|
| MacDonald and Langridge-Monopolis (1984) | $T_f = 0.0176 V_{er}^{0.364}$ |
| Bureau of Reclamation (1988) | $T_f = 0.011 B_{avg}$ |
| Froehlich (2008) | $T_f = 63.2 \left( V_w / g H_b^2 \right)^{0.5}$ |
| Xu and Zhang (2009) | $T_f / T_r = 0.304 (H_d/H_r)^{0.707} \left( V_w^{(1/3)}/H_w \right)^{1.228} e^{B_5}$ |
| Soliman (2015) | $T_f = 0.15 + 1.865 H^{-0.675} V^{0.408}$ |
| Ashraf et al. (2018) | $T_f = 5.935 H^{-0.9499} V^{0.4135}$ |

**Table 6.** Formulas to calculate the peak outflow $Q_p$

| Reference | Equation |
|---|---|
| MacDonald and Langridge-Monopolis (1984) | $Q_p = 1.154 (V_w H_w)^{0.412}$ |
| Bureau of Reclamation (1988) | $Q_p = 19.1 H_w^{1.85}$ |
| Froehlich (1995b) | $Q_p = 0.607 V_w^{0.295} H_w^{1.24}$ |
| Xu and Zhang (2009) | $Q_p / \sqrt{g V_w^{5/3}} = 0.175 (H_d/H_r)^{0.199} \left( V_w^{(1/3)}/H_w \right)^{-1.274} e^{B_4}$ |
| Pierce et al. (2010) | $Q_p = 0.038 \left( V_w^{0.475} H_w^{1.09} \right)$ |
| Ashraf et al. (2018) | $Q_p = 127.3 H_d^{0.6313} V^{0.7637}$ |

obtained from the extrapolated rating curve, i.e. 980 $\mathrm{m^3 s^{-1}}$. Nothwithstanding this, the downstream outflow (boundary) is
relatively well-known and reliable. This offers the opportunity to look for the upper boundary inflows, given the additional
inputs of the Pliessnitz and Czerwona Woda rivers are also relatively well defined, but of smaller relative importance (maximum
discharge rates of 46 and 36 $\mathrm{m^3 s^{-1}}$, respectively).

The current two-dimensional modelling was executed using MIKE21 software (McCowan et al., 2001; DHI, 2011). The
following works contributed to the model creation: i) preparation of the digital elevation model (DEM) based on both Polish
and German data; ii) generation of the bathymetry of the main river channel based on field surveyed cross-sections, making use
of a linear interpolation; iii) generation of the calculation bathymetry with a regular grid resolution of 5m x 5 m, by merging
the main channel bathymetry with the DEM (in ArcGIS environment); iv) implementation of hydrotechnical structures and
buildings as well as linear structures (e.g. embankments) by adjusting the ordinates of corresponding grid cells. v) preparation
of the roughness raster from the land cover based on aerial photographs; vi) formulation of boundary conditions in a form
of water level and discharge series (IMGW, 2011). Since the Lusatian Neisse downstream channel was modeled as an open-
ended reach, the downstream boundary condition was set as a normal depth. The total number of grid cell was 2.65 million,
the computation step used was from 0.5 to 0.75 sec to achieve the proper Courant numbers and the numerical stability on the
one hand, and feasible computation times on the other hand.





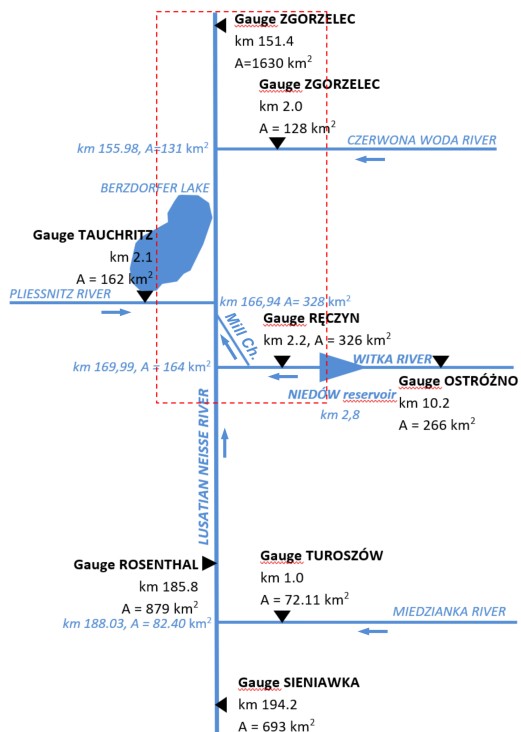

**Figure 9.** Hydrological scheme for the modelling domain (dashed frame indicates the 2D model area)

## 4    Results and Discussion

### 4.1    Dam breach formula performance

Calculated breach characteristics for the left and right embankment are collected in Table 7, 8 respectively. The results show relatively small differences, within ±10%, concerning the hight of the breach, which is a positive aspect. Also the volume of the eroded material from the embankments calculate with the formula by MacDonald and Langridge-Monopolis (1984) is close to the real values. However, the calculated breach width is significantly underestimated, even by 50% for the left embankment. The best result here is delivered by the formula of Xu and Zhang (2009), with a discrepancy of 27%. For the right breach, the formulas' results appear somewhat better, with an error in a range of -46 to +15%. The likely reason of these discrepancies, at least contributing to that, is the presence of the (two layers) concrete plates reinforcing the upstream dam slope, which hinders the breaching depth in favor of breaching width. Nevertheless, the largest discrepancies appear regarding the breaching time. Surprisingly, the two formulas (Froehlich, 2008; Xu and Zhang, 2009) overestimated this parameter even by ca. 130% and 40% for the left and right embankment, respectively. The other formulas yield an error ranging from -28% (MacDonald and Langridge-Monopolis, 1984) up to -60% (Bureau of Reclamation, 1988) for the left dam. For the right dam the error similarly negatively ranges from -59% (Ashraf et al., 2018) up to -70% (MacDonald and Langridge-Monopolis, 1984).





**Table 7.** Calculated left side breach characteristics for the Niedów dam (in brackets - percentage relative to the real life data)

| Reference | Avarage breach width $B_{avg}$ m | Volume of dam material eroded $V_{er}$ m$^3$ | Breach hight $H_b$ m | Breach time $T_f$ h |
|---|---|---|---|---|
| MacDonald and Langridge-Monopolis (1984) | 49.8 (-53%) | 27025 (8%) | - | 0.70 (-60%) |
| Bureau of Reclamation (1988) | 35.1 (-67%) | - | - | 1.29 (-28%) |
| Froehlich (2008, 1995b) | 56.1 (-47%) | - | - | 4.09 (130%) |
| Xu and Zhang (2009) | 77.4 (-27%) | - | 12.76 (10%) | 4.15 (133%) |
| Soliman (2015) | 56.3 (-47%) | - | 10.24 (-12%) | 0.87 (-51%) |
| Ashraf et al. (2018) | 57.4 (-46%) | - | 11.07 (-5%) | 1.17 (-34%) |
| Niedów dam left | 106.0 | 25130 | 11.6 | 1.78 |

The peak of total outflow through the spillway and two sections of breaching dam (individual peaks are shifted in time) is taken for comparison with the formulas results due to the statistical methodology used for their development. As presented in Table 9, also in this case there are significant discrepancies with the real life estimates. Most of the calculated peak outflows are overestimated, with an error in a range of 3% (Froehlich, 2008) to 122% (Ashraf et al., 2018). Only the formula by Pierce et al. (2010) delivered an underestimation of 20%.

**Table 8.** Calculated right side breach characteristics for the Niedów dam (in brackets - percentage relative to the real life data)

| Reference | Avarage breach width $B_{avg}$ m | Volume of dam material eroded $V_{er}$ m$^3$ | Breach hight $H_b$ m | Breach time $T_f$ h |
|---|---|---|---|---|
| MacDonald and Langridge-Monopolis (1984) | 31.4 (-46%) | 15993 (-23%) | - | 0.66 (-70%) |
| Bureau of Reclamation (1988) | 35.1 (-39%) | - | - | 0.63 (-71%) |
| Froehlich (2008, 1995b) | 45.1 (-22%) | - | - | 2.91 (35%) |
| Xu and Zhang (2009) | 66.6 (15%) | - | 12.76 (10%) | 3.19 (48%) |
| Soliman (2015) | 45.5 (-21%) | - | 10.05 (-13%) | 0.69 (-68%) |
| Ashraf et al. (2018) | 50.9 (-12%) | - | 10.97 (-5%) | 0.88 (-59%) |
| Niedów dam left | 58.0 | 20860 | 11.6 | 2.16 |

The origin of the discrepancies is attributable to specific features of the studied dam wash-out and may be listed as follows:

1. The inflow direction relative to the dam axis plays a relevant role.




**Table 9.** Calculated summary peak of discharge from the breach for the Niedów dam (in brackets - percentage relative to the real life data)

| Reference | Peak outflow rate $Q_p$ $\mathrm{m^3 s^{-1}}$ |
|---|---|
| MacDonald and Langridge-Monopolis (1984) | 2281 (68%) |
| Bureau of Reclamation (1988) | 1808 (33%) |
| Froehlich (2008, 1995b) | 1398 (3%) |
| Xu and Zhang (2009) | 2712 (100%) |
| Ashraf et al. (2018) | 3014 (122%) |
| Pierce et al. (2010) | 1088 (-20%) |
| Niedów dam left | 1356 |

2. The presence of the asphalt road on top of the dam and a relatively shallow water layer flowing over the dam crest, up to 0.45 m, caused a relatively long time of downstream slope washing-out and the appearance of breaching on the right side along with the significant breaching width. Typically, the earth dams breach grows rapidly from the very beginning.

3. After the dam crest collapsed, the resistance of the upstream concrete slabs hindered the growth of the breach down, thus it is accompanied with the growth of the width; this also may explain why the breach width on the left side was twice as large as on the right side.

4. The failure of the training wall contributed to the peak outflow.

5. A slowed down breach development for a small reservoir volume, resulted in lower outflow rates compared to most formulas.

6. The method by Xu and Zhang (2009) surprisingly does not perform well in the current case. Likely, there is an insufficient number of cases used for the non-linear regression analysis (5 cases), and a specific character of the Niedów dam failure plays its role.

In general, the use of prediction formulas, even though they are not complicated in nature, is not straightforward, demanding several assumptions and auxillary, sometimes approximate computations, irrespective of whether they are applied for real life cases or prediction trials. Usefulness of such approaches in specific configurations may be questionable, with limited reliability of outcomes, obviously requires caution in application and interpretation. It is clear that the reinforcements like concrete surface of the upstream slope or the dam crest, as in the studied case, is an important but omitted or not well tackled factor. The omission of this may lead to a serious over prediction of the dam break consequences, consequently to expanded mitigation costs. A sensitivity analysis should further be extended to assess the relevance of particular parameters on the formula outputs. This





study highlights the need for more work in this subject, especially relating the breaching dynamics with the physical properties
of the dam and hydraulic parameters in place of rather robust regression analysis. A unified methodology and general approach
is highly desirable.

## 4.2   2D Flood Routing

The flooding along the Lusatian Neisse in the studied reach is a combination of two major flood vawes (including the catastro-
phe of the Niedów dam) at the upper boundaries (see Fig. 10) with hydrographs to be reconstructed.

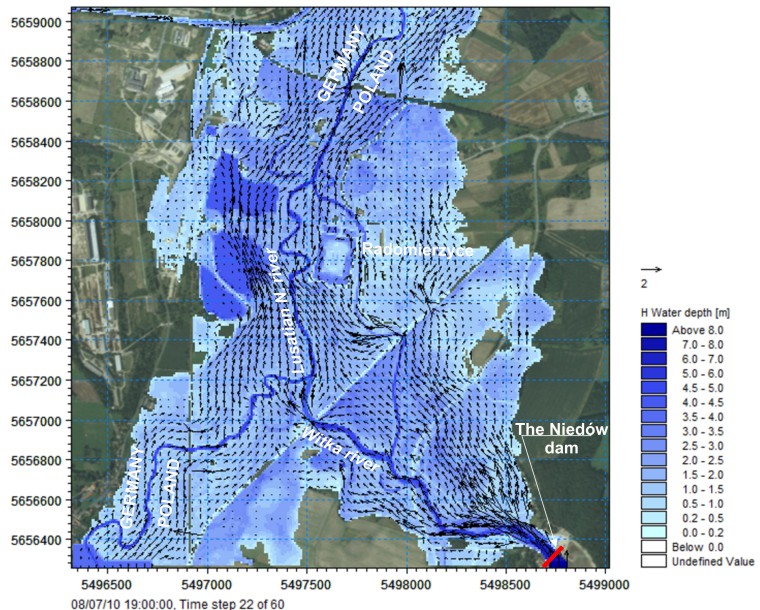

**Figure 10.** Simulated water depths and flow velocity vectors downstream the Niedów dam

The solution of the problem is iterative, relying on a series of computations executed with adjusted shapes of hydrographs,
timing and volume, conforming with Eq. 1. In this iterative approach, a simultaneous sensitivity analysis was performed on
identifying and analyzing the influences of change of the roughness Manning coefficients on peak flows and on wave front
propagation for a given breaching outflow hydrograph. The consequent calibration process also included local bathymetry
modifications of the main channel, which appeared necessary due to two reasons: i) rectangular grid is not always properly
representing relatively narrow and curved channels; ii) the cross-sections taken during low or medium flows may not be
representative for high flow conditions and local scouring needs to be taken into account (local roughness coefficients would
be beyond an acceptable range). Yet, one also needs to bear in mind that high water marks are not absolute values in terms of
accuracy, some of them are just indicative. The author's being a field surveyor and modeler in one person is an advantage here.
Fig. 11 further illustrates the flooding over the whole modelled area at 10:00 of 8 August 2010, when the flood peak reaches the
Zgorzelec city. This figure also indicates the location of high water marks (denoted as WW) spread on the right bank. The final





computation is considered as a consistent reconstruction of the flood in terms of water levels, flooding extend, discharge rates, timing and water volumes. The simulated water levels correspond well with the high water marks, given the complexity of the modelled domain and the inherent uncertainties; the differences are from just a few to about thirty centimeters. Remarkably, the 2D model delivered a close to reality inflow to the Berzdorfer lake due to embankment overtopping. The resulting dam breach

hydrograph $Q_N(t)$ was determined with a peak discharge of 1380 $\mathrm{m^3s^{-1}}$ appearing at 18:20, see Fig. 11. The total volume of water released due to the dam failure is 22 million $\mathrm{m^3}$, which is about 5 mln $\mathrm{m^3}$ more than the inflow to the reservoir.

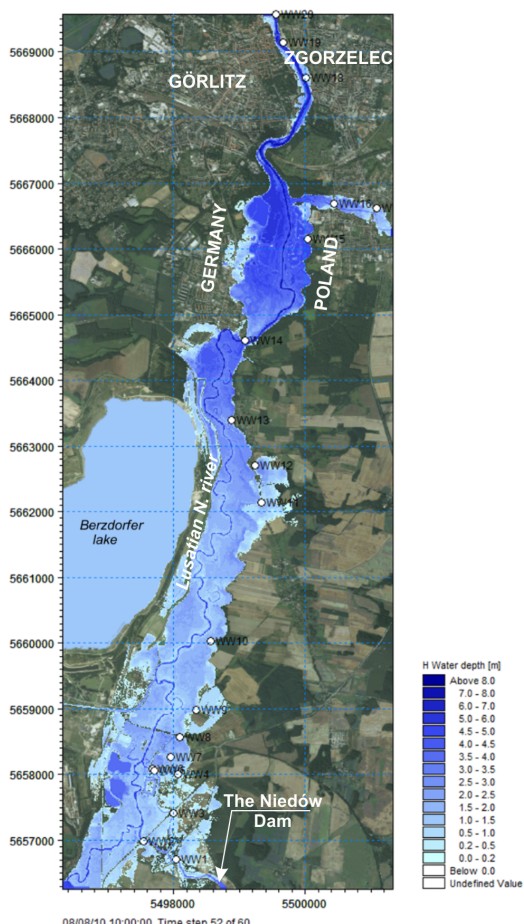

**Figure 11.** 2D simulation of the flood on the Lusatian Neisse in 2010 (WW - high water marks)

Fig. 13 presents the crucial discharge hydrographs reflecting the flood wave transformation along the river. The influence of valley retention on the flood propagation is remarkable. This retention was of about 20 mln $\mathrm{m^3}$, not counting for the Berzdorfer lake. Owing to that, there was a significant reduction of the flood peak discharge from 1730 $\mathrm{m^3s^{-1}}$ at the cross-section near

the Witka mouth (km 169.5) to 950 $\mathrm{m^3s^{-1}}$ at the Zgorzelec gauge station (km 151.4), hence preventing larger damages in the Zgorzelec city. The Lusatian Neisse hydrograph shows two peaks – the first one is caused by the Niedów dam break, and





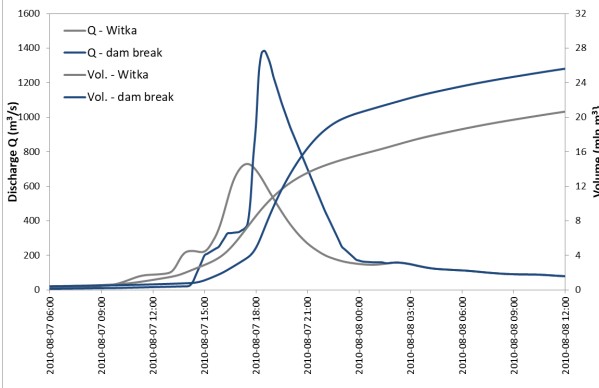

**Figure 12.** Hydrographs of discharge as the inflow to the reservoir and the outflow as the result of dam break determined in the course of iterative 2D modelling. Secondary axis presents the cumulated outflow volume - the final difference corresponds to the reservoir storage at normal conditions

the second one by the flood of the Lusatian Neisse, which culminated, fortunately, about nine hours later relative to that of the Witka river. As depicted in Fig. 13, the travel time of the first flood peak, originating from the outflow from the Niedów reservoir, to the Zgorzelec gauge station was about seven hours while the second peak of the Lusatian Neisse traveled about

4.5 hours, which is reasonable, taking into account the fact that the second peak travelled over already inundated area.

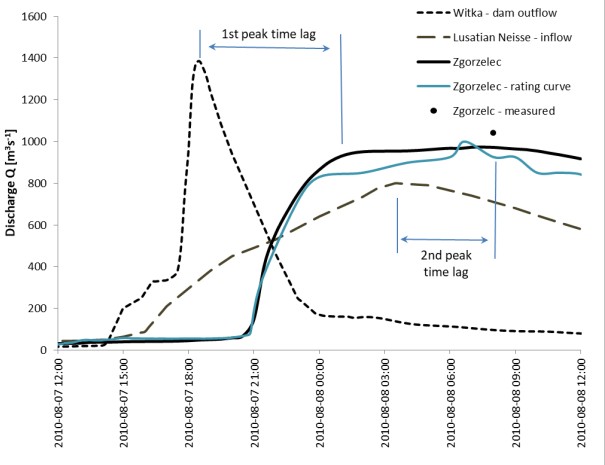

**Figure 13.** Hydrographs of discharge for the Witka and the Lusatian Neisse river





## 5    Conclusions

The literature review and the current case study demonstrate that the dam breach mechanism and prediction is an extensively studied and complex subject. There is a variety of failure modes and possible approaches to quantitatively assess the dam break dynamics and consequences. So, in this context the study has undertaken an effort to reconstruct and explain the catastrophic

event of the Niedów dam failure to contribute to the current data base on the dam breach development.

Hence, a detailed dam breaching mechanism is provided along with the final breach parameters that can be used for statistical analyses or physically-based model development. A particular feature of the Niedów dam was that it consisted of a concrete spillway, an integrated powerhouse intake structure, and two flanking earth embankments. The embankments were homogenous sand and gravel, with a concrete facing as the impermeable barrier and the asphalt road on the top. That substantially affected

the process of two-sides embankment washing-out, which appeared significantly different and longer than for uniform earth embankments. Consequently, the paper shows the inadequacy of the most prediction formulas for the dam breach of the current case. This is an important example as misleading results of dam breach prediction in respect to the outflow hydrographs, in particular peak discharge, can have serious consequences on the flood risk assessment and mitigation. Therefore, every case study must be considered with caution. The paper also presents a successful implementation of a 2D hydrodynamic model

for the simulation of flood wave propagation with an reversed solution of the upper boundary hydrographs for a compound flood event, including the one of the Niedów dam failure. This modelling approach is considered as an alternative to the outflow hydrograph assessments based on the statistical formulas. Remarkably, in the course of extensive modelling a large area reconstruction of the flooding was possible in data limited situation, leading to important answers to stakeholders in a bilateral, transboundary context.

*Author contributions.*  SK and RB collected the data, conceived the study, interpreted the results and wrote the paper. SK assessed the dam break formula performance while RB conducted the flood routing. The authors revised and approved the paper.

*Competing interests.*  The authors declare that they have no conflict of interest.



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
