# Peer review of "The catastrophe of the Niedów dam – the dam break causes, development and consequences"

_Natural Hazards and Earth System Sciences, 2020_

## Referee Comment (RC1) · Anonymous Referee #1 · 29 Jan 2021

This paper documents an important dam breach event (Niedów dam-breach) that occurred in 2010 during an extreme synoptic rainfall event that involved a large watershed. The test case is interesting although it could be better documented. Regarding the methods, some empirical equations are used to model the dam breach and a widely used 2D hydrodynamic model is used for the simulation of the flood wave in the long stretch of the river downstream of the dam. In spite of the interest of the test case, the paper is not well written and it is not clear what are its scientific reasons of interest. The inadequacy of empirical dam breach formulas to represent such a complex situation is not a surprise and the meaning of the claim of "propagation with an (sic) reversed solution of the upper boundary hydrographs" is not clearly understandable, because too

many points in the paper are presented in a rather confusing and non-reproducible way. Much of the reasoning is based on Eq. 1 that however is a wrong transient mass balance equation because it disregards storage in the floodplain flooded area, which must be relevant in the particular case. However, no variation of the stored water volume appears in this equation, where inflow hydrographs equate, at each time step, the output hydrograph. Finally, even without being an English mother tongue, I would say without fail that this paper will certainly benefit from a careful and overall improvement of the text that, as it is, is not suitable for an international Journal. In the following (see attached list) I provide a wide set of examples but many more are still present throughout the paper.

Accordingly, in my opinion, in its current form the paper is not suitable for publication.

Apart from these considerations, in many places the description of the different issues is involved and logically wrong (e.g., see lines 85 and 87 to better understand what I mean). As I said the test case is interesting and I invite the Authors to write a better paper to document it. Apart from the fundamental issue of scientific correctness and methodological clarity, it could be better documented for instance with pictures showing the different phases of the event, maybe provided as additional material. From a technical point of view, an undiscussed aspect is why they waited so long to open the gates (see Table 1 progression of the opening of gates I,II and III) As a final note, I would recommend using the term dam breach throughout the paper in place of dam break, a term that in the field of hydraulics is related to the impulsive collapse of a dam, as typical for reinforced concrete or masonry structures.

I enclose a pdf file with a detailed list of my observations

Please also note the supplement to this comment:
https://nhess.copernicus.org/preprints/nhess-2020-372/nhess-2020-372-RC1-supplement.pdf

[Figure]

**Supplement:**

| Line | | Observation |
|---|---|---|
| 64 | Solutions | Do you mean analytical solutions ? I doubt. Accordingly I would say "physically based models" |
| 76 | ANN model | What is it ? Acronyms should be defined the first time they are used |
| 83 | Such a construction is likely to significantly affect t | Only in case of overtopping, I suppose, but not in case of siphoning. |
| 85 | The evaluation of the risk of failure and its consequences relies on testing of a number of the catastrophe scenarios | The risk of failure is not based on the catastrophe scenarios, which is, rather, a consequence of the failure. |
| 87 | modelling of …flood plain flows as well as inundation maps | Inundation maps are not an additional item but are the results of modeling. |
| 89 | In channels | Do you mean in narrow valley ? This is true and a clear example is the often cited paper by Pilotti et al, ASCE, 2011. But in general a 2D approach in floodplain is mandatory. So I would cancel the reference to hybrid 1D/2D model. |
| 92 | Although 1D models are most used … | I would delete this phrase as far as line 94 because it contains a repetition and the following is not the state of the art. |
| 99 | along with determination | and for the determination |
| 115 | a rainfall sum of two days | A cumulated 48-hours rainfall |
| 116 | , see Fig. 2 | (see Fig. 2). |
| 128-130 | There are two stations… | It would be useful to plot the discharge hydrograph at the two stations as far as available |
| Figure 1 | | Improve: add numbers indicating Witka river, the Miedzianka river and Lusatian Neisse river which are mentioned in the paper but missing in the Figure
Moreover, no indication appears here about the area studied in the 2D modeling, that pops up only in Figure 9 |
| 152 | This information helps explaining … | Not clear. Do you mean that the maximum water elevation occurred due to the coincidence of two separate flood waves and in corrsepondence of the peak of the dam breach wave ? |
| Figure 3. | | Add major dimensions on the blueprint and dashed lines to show the location of the cross section shown in Fig. 4 and 5 |
| 167 | The body of the dam was well compacted sand | Only sand with no clay core present ? Is this a common practice or something that deserve some additional comments ? |

| 176 | An S-shape plan of the reservoir just upstream the dam (Fig. 1) | How can one see it ? Add in Fig. 1 an enlarged map of the reservoir with the dam position
The area in which the reservoir is located is not entirely clear to an audience which does not know a priori the area under investigation. A more detailed or enlarged map of the reservoir is needed to help the reader visualize the local geographical context. Furthermore, in my opinion, since the geometrical characteristics of the reservoir play an important role in the article, it is mandatory to show such properties in a map containing the main dimensions involved. |
|---|---|---|
| | caused a low angle of inflow direction to the dam axis. As a result, the water level at the left side was higher several centimetres than on the other side, leading to uneven overflow | Without a better layout description this statement sounds not realistic. What evidence do you have for this ? I guess that the kinetic component in a reservoir is negligible and the radius of curvature very large and, accordingly, the bend superelevation in a curve must be undetectable.  Here, as in several other points of the paper, the description is to vague and unprecise for the reader to really understand what process is in action. For sure an enlarged map of the reservoir upstream of the dam is needed. |
| 178 | The left dam | Do you mean the left side of the dam ? |
| 179 | lighting foundations 180 (additional turbulences) | What is it ? What do you mean ? |
| 188 | the earth dam was almost completely swept a | the left side of the earth dam |
| 197 | The final width of this breach was 58 m as illustrated by Fig. 8. | From the figure one has not this piece of information. Is it the sum of the left and of the right breach widths ? |
| Table 1 | At 17:42 dam – 40 m on the left side and 30 m on the righ | What do you mean ? Is it water submergence in cm ? |
| 226 | . The growing width of the breach was interpolated based on the photographs and films made during the catastrophe | These documents should be ordered and made available as additional material. |
| 225 | For the dam breaches, first a broad crested weir formul | This part provides the computed discharge. Apart that a steady state equation is used to represent a transient phenomena and that two parallel weirs are in action at the same time, no data (e.g., discharge coefficient….)  are provided to really understand how the computation was accomplished . Moreover the evaluation of the breach height |

| | | |
|---|---|---|
| | | using the 2D model is badly explained and potentially totally arbitrary. |
| 239 | Properly restore | Explain better. What do you mean ? |
| 243 | Eq (1), | This mass balance equation is wrong because it disregards storage in the flooded area. In a following part of the paper we are informed that (see line 333) "The influence of valley retention on the flood propagation is remarkable. This retention was of about 20 mln m3, not counting for the Berzdorfer lake." However, no variation of the stored water volume appears in this equation, where inflow hydrographs equate, at each time step, the output hydrograph.

Moreover Eq (1) shows the mass balance introduced to solve "iteratively" the unknowns present in the numerical model. In the following lines only the term $Q_{ND}(t)$ is classified as unknown and the other terms being known by the authors at some time (the discharge into the lake should be computed automatically by MIKE 21 using the floodplain topography), so what exactly is the iterative procedure used in the numerical model is not clear. |
| 265 | Since the Lusatian Neisse downstream channel was modeled as an openended reach, the downstream boundary condition was set as a normal depth. | I suppose that the flow is subcritical at this cross section and you have a stage-discharge relationship that was further enriched by a measurement at the peak of the event at Zgorzelec cross–section
Why didn't you use the measured stage-diacharge curve, Q(h), as a downstream boundary condition in place of a normal depth that could be unjustified ? Actually how can you be sure that you have not any backwater effect from downstream at the Zgorzelec cross–section ? |
| 264 | preparation of the roughness raster from the land cover based on aerial photographs; | Detail better. Provide a map with roughness coefficient. |
| 268 | proper Courant numbers | What is the proper value in your case ? |
| 285 | the real life estimates | Real life ? Do you mean the values computed using the weir equation at line 225 ? |
| 285 | The origin of the discrepancies is attributable to specif | Why do you not consider that a major discrepancy arises from the complex layout of the breach in this case ? Actually in your case you have two parallel and indipendent breaches developing at the same time. I doubt that any of the empirical equation considered makes explicit reference to such a complex situation. |
| Figure 10 | | Add dimension beside 2 for the velocity vector. This velocity map is very hardly understandable due to the overlapping of |

| | | the vectors. Please add a second map with color shading for velocity only. |
|---|---|---|
| 293 | | The sentence reported is vague and does not provide a complete explanation of the reason of the difference between the left and right embankment in terms of the breach width, which, as stated in the current work, is very important in all the reported calculations. A more precise explanation is needed in my opinion. |
| 315 | The solution of the problem is iterative | This is not clear. The problem is not clearly set |
| 320 | local scouring needs to be taken into account (local roughness coefficients would be beyond an acceptable range). | These are two separate issues: the fixed bed hypothesis and the uncertainty of the local roughness. What do you mean with this statement ? |
| 322 | . Yet, one also needs to bear in mind that high water marks are not absolute values in terms of accuracy, some of them are just indicative | Again, a phrase without a proper explanation. Clearly, any measurement is affected by uncertainty but what do you exactly mean ? |
| 325 | high water marks | Is there a plot that compare measurements and modelled water elevations ? |
| 329 | the 2D model delivered a close to reality inflow to the Berzdorfer lake | On what basis one can conclude that the inflow is close to reality ? |
| 330 | The resulting dam breach hydrograph QN (t) was determined with a peak discharge of 1380 …at Fig. 11 | The Figure is 12 I suppose. |
| | Figure 12 | I do not understand the physical reason of the discharge plateau in the outflowing dam breach hydrograph of Figure 12. If the Witka entering discharge is growing in time and the outflowing discharge is constant, it means that the level in the reservoir is growing. Why the outflowing dam breach hydrograph does not grow with the water level in the reservoir ? |

| Figure 9: | | In the figure there are red marks underneath the names of the locations reported typical of a grammatical check tool, their presence is unessential. Furthermore the schematics provided is very helpful in understanding the domain in which the numerical method is applied, but the lack of a geografical counterpart of the same scheme (a map with all the locations highlighted) damages the understanding of the spatial dimensions involved in the simulation. As previously noted, a map is required to help the reader to orient himself in the various locations described in the current work. |
| --- | --- | --- |
| Figure 12 | | Colors used to depict the two hydrographs are too similar and generate confusion |

| Line | | Observation regarding questionable use of words |
| --- | --- | --- |
| 16 | In the twenty-first century, | Do you mean "in the twentieth century" |
| 19 | Ageing | Aging is better, as at line 23 |
| 33 | extend | extent |
| 35 | Very high | Do you mean "fast " ? |
| 37 | respective | on |
| 38 | development | evolution |
| 44 | Upon this… | Reword more properly |
| 48 | Estimated parameters | Not clear: do you mean "uncertain parameters" ? |
| 50 | | the breach height,  width, and side slope ratio |
| 53 | Accurate, | Delete comma |
| 55 | Dam forming material | dam, simply |
| 59 | and found out that the DLBreach model was found to be the most accurate | and found out that the DLBreach model was the most accurate |

| 67 | Fundamental for the determination of the outflow hydrograph is the breaching time | the breaching time is fundamental ... and then "which" used twice. Reword the phrase |
|---|---|---|
| 72 | the breach deformation time | "Chinnarasri et al. (2004) ..."; deformation ??? |
| 85 | catastrop | catastrophic scenarios in ..." |
| 99 | Finally, a two-dimensional (2D) hydrodynamic model ... | Reword the whole phrase |
| 103 | Cartographic Information | Description of the study area |
| 108 | was that of 183 hectares | was 183 hectares |
| 109 | significant head water slopes | ??? |
| 121 | appearance | occurrence |
| 134 | To discussion | significant uncertainty because a direct reliable estimation ... |
| 135 | In addition, the topography makes it more difficult due to | What do you mean ? reword this phrase |
| 146 | threes | trees |
| 189 | begn | The overtopping of the right dam began approximately .. |
| .... | | |
| 355 | an reversed | An inverse ? a reverse ? |
| | | |

---

## Short Comment (SC1) · 10 Feb 2021

A very interesting and well written article. A very good and detailed description of the breach mechanism. Taking the time to understand and evaluate dam failures, the breach mechanism and breach parameters, and comparing them to numerical models, as was done in this article, is critical work for dam safety. The better the modeling gets, the better job we can do in preparing of Emergency Action Plans, which are essential in protecting the public from potential disasters such as the Niedow Dam failure.

---

## Author Comment (AC1) · 12 Mar 2021

**Reviewers #1**

**The authors highly appreciate the comments of the Reviewer, helping to improve the paper quality. The authors carefully went through all comments and incorporated relevant amendments in the paper. Below are also responses to issues raised. First the general comments are addressed, further the specific ones.**

**General comments**

*RC. This paper documents an important dam breach event (Niedów dam-breach) that occurred in 2010 during an extreme synoptic rainfall event that involved a large watershed. The test case is interesting although it could be better documented.*

Thank you for this general and positive comment.

*RC. Regarding the methods, some empirical equations are used to model the dam breach and a widely used 2D hydrodynamic model is used for the simulation of the flood wave in the long stretch of the river downstream of the dam. In spite of the interest of the test case, the paper is not well written and it is not clear what are its scientific reasons of interest.*

We will clarify a number of issues following the specific comments. Concerning the scientific goals, they are:

To document and explain the causes of the dam breach of the Niedów dam as a case study; further to test several available empirical formula on the dam breach prediction, and finally to execute a numerical flood routing using a 2D model to determine the breach outflow hydrograph along with the flood propagation, relevant for the evaluation of the flood damage causes. A change is made in the introduction (L. 95-101):

The current work presents a case study of a catastrophic failure of the Niedów dam in Poland, in 2010. The goals of the study are to document and provide a detailed picture of the dam breach along with explanation of the failure mechanism in the case of an earth dam. This is made on the basis of own field survey, witnessed, stored, or restored data. The geographic, meteo- and hydrological conditions leading to this event are presented as well. The work further puts under the test several selected formulas with an attempt to assess the dam breach characteristics and the peak outflow. Finally, a two-dimensional (2D) hydrodynamic model is applied to more reliably determine the dam break outflow hydrograph and consequent flood wave propagation, in particular, quantifying the effect of the Niedów dam failure on the flooding downstream area.

*The inadequacy of empirical dam breach formulas to represent such a complex situation is not a surprise…*

The current dam breach development is specific, different from usually assumed schemes. In particular, the presence of the two-layered concrete slabs on the upstream dam slope as the impervious layer (seepage protection), combined with an asphalt road on top of the dam, hindered the breaching depth due to jamming of the slabs, with a consequence of relatively low outflow peak. On the other hand, the breaching width is larger, leading to different geometrical proportions of the breach, compared to those anticipated by the formula. This indicates that the breach forecast needs

to carefully take into account the dam structure and search for a most suitable dam breach forecast method.

*…and the meaning of the claim of "propagation with an (sic) reversed solution of the upper boundary hydrographs" is not clearly understandable, because too many points in the paper are presented in a rather confusing and non-reproducible way.*

We have made multiple changes to a number of statements, including the description of the modelling approach, accordingly to specific comments. By 'reversed solution' it was meant that unknown upper boundary inputs were sought based on known low (output) boundary. For this an iterative search was executed. Nevertheless, the word 'reversed' is removed for not being pretentious.

*Much of the reasoning is based on Eq. 1 that however is a wrong transient mass balance equation because it disregards storage in the floodplain flooded area, which must be relevant in the particular case. However, no variation of the stored water volume appears in this equation, where inflow hydrographs equate, at each time step, the output hydrograph.*

Eq. 1. was written as problem specific for the unsteady flow simulation. The term for the valley retention was originally not added as the water mass conservation is inherent to the 2D model. Nevertheless, for formal reasons and following Reviewer's comment we will add a term for change of the retention in the river valley, *-dV(t)*.

In fact, the mass conservation was of prior importance of this flood routing (therefore a 2D model was applied after an initial 1D attempt), to ensure that the calculated low boundary discharge hydrograph is conform with the gauge recordings.

*From a technical point of view, an undiscussed aspect is why they waited so long to open the gates (see Table 1 progression of the opening of gates I,II and III)*

To control the water level in the reservoir, the crew initially followed the operational manual. The procedure was to elevate stepwise the gate by 0,2 m to maintain the desired water level. If necessary additional gate was elevated by 0,2 m too. In the course of this unpreceded water level rise the gate opening was accelerated, nevertheless not to avoid of flooding of the control room. Finally the crew tried to open more the gates manually from the dam crest. The flooding occurred on Sunday, and the crew was not fully aware on the dynamics and scale of escalating flood damage on the Czech Republic territory, including the damage of the Frýdland gauge station on the Smeda/Witka river upstream. In fact, this dam operation was a subject of prosecution and the court trial with judgment acquitting all the persons charged.

*As a final note, I would recommend using the term dam breach throughout the paper in place of dam break, a term that in the field of hydraulics is related to the impulsive collapse of a dam, as typical for reinforced concrete or masonry structures.*

We use the term 'dam breach'.

*Finally, even without being an English mother tongue, I would say without fail that this paper will certainly benefit from a careful and overall improvement of the text that, as it is, is not suitable for an*

*international Journal. In the following (see attached list) I provide a wide set of examples but many more are still present throughout the paper.*

The paper before submission was checked by a "very experienced translator with an English knowledge on level Native" from a translating office. Even the Authors feeling was that the work was not done very well…

English will be improved and checked by another native speaker familiar with technical and environmental topics.

**Specific comments**

**Authors are grateful to the Reviewer for all numerous specific comments done. We took into account all of them, and here are the explanations (many corrections done in the text are not detailed here).**

L. 64. *Do you mean analytical solutions ? I doubt. Accordingly I would say "physically based models"* Changed into: physically based models

L. 76. ANN: changed into: artificial neural networks models

L. 83. *Only in case of overtopping, I suppose, but not in case of siphoning.*

The sentence is modified: Such a construction, in case of overtopping, is likely to significantly affect the breach time and breach geometry, hence it affects the outflow hydrogram and peak.

L. 85-94. Modelling. The whole paragraph is corrected:

The evaluation of the dam failure consequences relies on testing of a number of catastrophic scenarios to further analyze and assess the consequences of the potential flood. The bases of such analysis are hydrologic simulations, numerical modelling of breaching processes, and flood plain flows as well as preparation of inundation maps using GIS systems (Altinakar, 2008; Cleary et al., 2012, 2015; Cannata and Marzocchi, 2012; Álvarez, 2017; Zhong, 2011). 1D models can predict the flood propagation in channels and narrow valleys with reasonable accuracy and good efficiency (Pilotti, 2011), but a 2D or hybrid1D/2D approach should be used in wide floodplains and complex terrain regions with elevated roads, secondary dikes, levees, buildings, and other obstacles (Vanderkimpen and Peeters, 2008). The 2D models gained applications due to significant computational power advances and air-born topographical data availability in recent years (Saberi et al., 2013; Yakti et al., 2018, Banasiak, 2021). In addition, hybrid models are used to derive 1D based breach outflow hydrographs, whereas the 2D model is used for flood plain modelling and generation of inundation maps downstream the dam (Shah et al., 2019).

L.99, 115,116 corrected.

L. 128-130 – *It would be useful to plot the discharge hydrograph at the two stations as far as available*

Drawing the discharge hydrograph was highly desirable but problematic for both stations given the high flow magnitude respective to the measured range and the local topography. There are water levels hydrographs available, but the Ręczyn gauge was destroyed soon after the dam breach started. The determination of the peak discharge for the Ostróżno station was difficult as mentioned in the paper, because of the lack of rating curve for the unpreceded high flow in a wide floodplain.

Fig. 1. *Improve: add numbers indicating Witka river, the Miedzianka river and Lusatian Neisse river which are mentioned in the paper but missing in the Figure Moreover, no indication appears here about the area studied in the 2D modeling, that pops up only in Figure 9*

The figure is modified, including an enlarged plan view of the reservoir. Additional figure showing the Lusatian Neisse river catchment and river network is provided:

[Figure]

Fig. 1. The catchment of the Witka River

[Figure]

Fig. 2: The upper Lusatian Neisse catchment up to the Görlitz(DE)/Zgorzelec(PL) gauge station (source: IMGW, 2010).

L.152. *Not clear. Do you mean that the maximum water elevation occurred due to the coincidence of two separate flood waves and in correspondence of the peak of the dam breach wave ?*

The sentence is modified:

This information helps to reconstruct the flood wave hydrograph, quantify the argued effect of coincidence flood waves from the two rivers as well as to define upper boundary conditions for the hydrodynamic model.

Figure 3. *Add major dimensions on the blueprint and dashed lines to show the location of the cross section shown in Fig. 4 and 5*

Fig 3. s adapted as below:

[Figure]

The Witka river →

Stilling basin

B

A

Concrete layer

Electropower
and pumping station

Control structure

A

The Witka reservoir

Concrete layer

L. 167. *Only sand with no clay core present ? Is this a common practice or something that deserve some additional comments ?*

The body of the dam was well compacted sand without a clay core. To protect against water filtration the upstream dam slope was covered by two layers of concrete slabs with a bitumen sealing. cf. Fig. 5.

L. 176-178. *Without a better layout description this statement sounds not realistic. What evidence do you have for this ? I guess that the kinetic component in a reservoir is negligible and the radius of curvature very large and, accordingly, the bend superelevation in a curve must be undetectable. Here, as in several other points of the paper, the description is to vague and unprecise for the reader to really understand what process is in action. For sure an enlarged map of the reservoir upstream of the dam is needed.*

We have to agree with the reviewer's opinion that the kinematic component in this case can be undetectable because of a relatively large cross-section area upstream of the dam, so this claim has been removed from the paper. We tried to find out the reason for overflow first on the left dam side, even also considering a side overflow effect but this is still speculative. An additional 2D modelling for the reservoir area could provide more detail.  Nevertheless we reviewed the available data on the dam crest elevation,  and concluded that the uneven elevation of the dam crest is the primary reason for the start of the overflow. In addition, a plan view of the reservoir is enlarged in Fig. 1.

L.179. *'lighting foundations 180'.  What is it ? What do you mean ?*

Corrected to: "around the lamp post foundations"

L. 188. Right - the left side of the earth dam

L.197. *From the figure one has not this piece of information. Is it the sum of the left and of the right breach widths ?*

The sentence is modified: The final width of the breach of the right dam was 58 m. Fig. 8 illustrates the complete dam breach.

Table 1. *At 17:42 dam – 40 m on the left side and 30 m on the right - What do you mean ?*

It was meant the length of the collapsed road on the top of the dam.
Corrected to: 17:42: Washing-out of the lee side of the dam, destruction of the road on top of the dam – the dam breach width of 40 m on the left side and 30 m on the right side.

L.225 *This part provides the computed discharge. Apart that a steady state equation is used to represent a transient phenomena and that two parallel weirs are in action at the same time, no data (e.g., discharge coefficient….) are provided to really understand how the computation was accomplished . Moreover the evaluation of the breach height using the 2D model is badly explained and potentially totally arbitrary.*

Since the breach for the left and right embankment developed differently, it was necessary to calculate or estimate the input parameters, i.e. the reservoir storage at the crest level and water volume above the breach bottom at the time of failure regarding both embankments. The time-varying outflow of the control structure (all three tainter gates) and the two breach outflows were calculated separately using a well-known hydraulic formula. To calculate the outflow through the tainter gates, the discharge coefficients used are in accordance with those obtained from the laboratory physical model testing for the Niedów dam (Herrera-Granados and Kostecki, 2016). For the dam breaches, first a broad crested weir formula was applied until the road on the dam crest collapses, with a discharge coefficient $C=2/3 \cdot C_d \cdot (2g)^{0.5} = 1.70$. Next, a formula for the sharp crest weir inclined to the horizontal at an angle of 18.4 degrees (as the inclination of the concrete slabs) were adopted from Shesha Prakash et al., 2011, and finally a formula for no crest weir for the situation of breaching to the ground floor were used, with a discharge coefficient C= 1.70, according to USBR, 1987. The growing width of the breach was interpolated based on the photographs and films made during the catastrophe. The breach depth in specific moments, being the most difficult parameter to reconstruct, was calculated making use of the discharge-water level relationship for a section closely downstream the dam. This relationship was obtained in the course of the 2D hydrodynamic modelling. The breach depth was searched as to match the resulting outflow discharge to the discharge obtained in the course of 2D modelling. In case of differences  the breach depth was

adapted. Finally, based on the outflow volumes of the left and right breach values of *V* and $V_w$ are estimated. The following results and comments are presented in Section 4.

In fact, use of the formulas to asses all the dam breach parameters is not simple and need several assumptions with inherent uncertainty as discussed later on in the paper.

L. 226. *These documents should be ordered and made available as additional material.*

We considered providing additional material (photos and a video) as well, but the authorship issues are related which are quite difficult to be resolved.

L. 243. *This mass balance equation is wrong because it disregards storage in the flooded area.* …

Eq. 1. was written as problem specific for the unsteady flow simulation. The term for the valley retention was not added as the water mass conservation is an inherent feature of this 2D model. Water flowing into the Berzdorfer Lake is 'lost' from the flood routing domain, as it does not enter back the river valley. Nevertheless, for formal reasons and following Reviewer comment we add a term for change of the retention in the river valley, *dV(t)*. Eq. 1 is now:

$$Q_{NL,in}\left(t\right)+Q_{ND}\left(t\right)+Q_{P}\left(t\right)+Q_{CW}\left(t\right)-V_{B}-dV\left(t\right)=Q_{NL,Z}\left(t\right) \tag{0.1}$$

*Moreover Eq (1) shows the mass balance introduced to* solve "iteratively" *the unknowns present in the numerical model. In the following lines only the term* $Q_{ND}(t)$ *is classified as unknown and the other terms being known by the authors at some time (the discharge into the lake should be computed automatically by MIKE 21 using the floodplain topography), so what exactly is the iterative procedure used in the numerical model is not clear.*

The iterative procedure included the adaptation of the hydrograph shape for both upstream boundaries, i.e. for the Lusatian Neisse and the Witka river/outflow from the Niedów dam, as well as finding proper roughness coefficients to obtain a good match between computed and observed hydrographs at Zgorzelec cross-section (timing and peak flow rate). In addition, the total overflow to the Berzdorfer lake had to be ensured conform to data provided by the German party. This was quite complex and, in fact, consecutive iterations were numerous and time consuming taking into account the computer effort of two days for a single run at that time (2011).

L. 264 *Detail better. Provide a map with roughness coefficient.*

Both the bathymetry and roughness raster (here: the velocity coefficient) will be added as supplementary files. The description now is:

v) preparation of the initial roughness raster from the land cover based on aerial photographs; (in total 15 roughness classes were distinguished – for the main channel and open surface waters, grassland and tree areas, bushes, paved surfaces, roads, etc.);

L.265. *I suppose that the flow is subcritical at this cross section and you have a stage-discharge relationship that was further enriched by a measurement at the peak of the event at Zgorzelec cross–*

*section Why didn't you use the measured stage-discharge curve, Q(h), as a downstream boundary condition in place of a normal depth that could be unjustified ? Actually how can you be sure that you have not any backwater effect from downstream at the Zgorzelec cross–section ?*

MIKE21 v.2010 at hand was not equipped with a stage-discharge relationship closing boundary. Instead, the water elevation in a function of time was used. This is simply the hydrograph of the Zgorzelec gauge station, as the modelled area ends in its cross-section.

The text reads now:

Since the Lusatian Neisse was modeled as an open-ended reach, the downstream boundary condition was set as the water elevation in a function of time. This was based on the Zgorzelec gauge observations directly as the modelled area ends in its cross-section.

L. 268. *What is the proper value of Courant number in your case?*

Text amendment made: The size of modelled area was 13.3 km by 5.0 km and the total number of grid cells was 2.65 million, the computation step was from 0.5 to 0.75 sec. to limit the Courant number to a value of one (although MIKE21 is capable dealing with larger values, DHI, 2011) and to achieve numerical stability and accuracy on the one hand, and feasible computation time on the other hand.

L.285. *Real life ? Do you mean the values computed using the weir equation at line 225 ?*

Modified to: As presented in Table 9, also in this case there are significant discrepancies with our case estimate of 1380 m3/s, based on the flood routing.

L.285. *Why do you not consider that a major discrepancy arises from the complex layout of the breach in this case ? Actually in your case you have two parallel and independent breaches developing at the same time. I doubt that any of the empirical equation considered makes explicit reference to such a complex situation.*

We agree with the Reviewer opinion. Indeed the breaching process under concern was compound and complex, especially the influence of concrete shield was important, hence the deviations between the calculated and assessed peak outflow (the last based on the flood routing) is not surprising. Nevertheless we consider this formula testing is worth presenting not only for their reliability assessment but also for the usage restrictions, as mentioned in the text. We used all formulas to calculated breach characteristics for both dam sides separately.

L. 293. *The sentence reported is vague and does not provide a complete explanation of the reason of the difference between the left and right embankment in terms of the breach width, which, as stated in the current work, is very important in all the reported calculations. A more precise explanation is needed in my opinion.*

Thank you for this indication again. We have corrected this to read:

After the dam crest collapsed, the upstream concrete slabs hindered the growth of the breach down, but less affected the breach horizontal development. The resistance of the concrete slabs may also

explain, along with a time delay of breaching inception, why the breach width on the right side was a half of that on the left side.

Fig. 9. The figure is corrected, the size of the modelled area is 13.3 by 5.0 km2 (2660 by 1000 cells 5x5 m2 each). Numbers are added to the text.

Fig. 10. The flow velocity unit is added. We tested several display setting in MIKEView (no color shading for velocity) and make unchanged the visualization of both water depth and velocity vectors at once as a compromise of content and clearance.

L. 315. *'The solution of the problem is iterative.' This is not clear. The problem is not clearly set*

The problem setting is now:

The flooding along the Lusatian Neisse in the studied case is a combination of two major flood waves originating from the upstream river section and the Niedów reservoir outflow. These are two upper boundaries of the modelled domain (see Fig. 10) with hydrographs to be reconstructed.

The reconstruction of these hydrographs was iterative, relying on a series of computations executed with adjusted shapes of hydrographs, including change of timing and rate of peak flows, to satisfy Eq. 1.

L. 320-322. *These are two separate issues: the fixed bed hypothesis and the uncertainty of the local roughness. What do you mean with this statement ? Again, a phrase without a proper explanation. Clearly, any measurement is affected by uncertainty but what do you exactly mean ?*

In fact this is an important issue. In other words, the model was not 'forced 'to be fully conform with several uncertain water level marks. Further, it can be discussed whether the fixed/eroded bed and roughness related uncertainty are two separate issues, or they are linked. To clarify, Lines 320-324 have been changed to:

ii) the cross-sections taken during low or medium flows may not be representative for high flow conditions. The local velocity coefficients M were kept in a range of 36 m1/3/s. This was the case in the section near the Zgorzelec city. Yet, one also needs to bear in mind that high water marks collected after the flood passage, against which the model is calibrated, are affected by (significant) uncertainty. Therefore, a balance on confidence between the model results and field data is sought. The author's being a field surveyor and modeler in one person is an advantage here

L. 325. *Is there a plot that compare measurements and modelled water elevations ?*

The water level hydrographs were drawn in each high water mark location, however here we limit the information to maximum values only. Table 10 is added: Comparison between estimated and calculated water levels.

Table 10. Comparison between estimated and calculated water levels

| Water marks | H (m) | H calculated (m) | Difference (m) |
| --- | --- | --- | --- |
| WW1 | 200.73 | 200.656 | -0.074 |
| WW2 | 199.04 | 199.121 | 0.081 |

| | | | |
|---|---|---|---|
| WW3 | 199.63 | 199.539 | -0.091 |
| WW4 | 197.72 | 197.618 | -0.102 |
| WW5 | 198.03 | 197.47 | -0.46 |
| WW6 | 197.66 | 197.382 | -0.278 |
| WW7 | 197.56 | 197.383 | -0.177 |
| WW8 | 197.22 | 197.327 | 0.107 |
| WW9 | 195.17 | 195.09 | -0.08 |
| WW10 | 193.91 | 193.942 | 0.032 |
| WW11 | 191.53 | 191.312 | -0.218 |
| WW12 | 191.36 | 191.125 | -0.235 |
| WW13 | 190.22 | 190.008 | -0.212 |
| WW14 | 189.09 | 189.203 | 0.113 |
| WW15 | 188.31 | 188.466 | 0.156 |
| WW16 | 188.53 | 188.463 | -0.067 |
| WW17 | 188.77 | 188.727 | -0.043 |
| WW18 | 185.43 | 185.498 | 0.068 |
| WW19 | 184.79 | 184.658 | -0.132 |
| WW20 | 182.77 | 182.754 | -0.0 |

L. 329. *On what basis one can conclude that the inflow is close to reality ?*

The text now: The calculated overflow volume to the Berzdorfer lake amounted 3,783 mln m$^3$, which is in accordance with the data provided by the German party, assessed based on the water level increase in the lake before and after the flood passage.

L. 330. All figure numbers are checked taking into account the additional one (Fig. 2).

Fig. 12. *I do not understand the physical reason of the discharge plateau in the outflowing dam breach hydrograph of Figure 12. If the Witka entering discharge is growing in time and the outflowing discharge is constant, it means that the level in the reservoir is growing. Why the outflowing dam breach hydrograph does not grow with the water level in the reservoir ?*

Figure 9  The indicated plateau is a results of the yield of water of the control structures. Please remind, that the outflow occurs underneath the gates, so the flow increase with water level is not that large as for free surface overflows.

*In the figure there are red marks underneath the names of the locations reported typical of a grammatical check tool, their presence is unessential. Furthermore the schematics provided is very helpful in understanding the domain in which the numerical method is applied, but the lack of a geographical counterpart of the same scheme (a map with all the locations highlighted) damages the understanding of the spatial dimensions involved in the simulation. As previously noted, a map is required to help the reader to orient himself in the various locations described in the current work.*

Figure 9  is corrected (below) and a catchment map is added as Fig 2.

[Figure]

**Gauge ZGORZELEC**
km 151.4
A=1630 km²

**Gauge ZGORZELEC**
km 2.0
A = 128 km²

*CZERWONA WODA RIVER*

*km 155.98, A=131 km²*

*BERZDORFER LAKE*

**Gauge TAUCHRITZ**
km 2.1
A = 162 km²

*PLIESSNITZ RIVER*

*km 166,94 A= 328 km²*

**Gauge RĘCZYN**
km 2.2, A = 326 km²

*Mill Ch.*

*km 169,99, A = 164 km²*

*WITKA RIVER*

**Gauge OSTRÓŻNO**
km 10.2
A = 266 km²

*NIEDÓW reservoir*
*km 2,8*

*LUSATIAN NEISSE RIVER*

**Gauge ROSENTHAL**
km 185.8
A = 879 km²
*km 188.03, A = 82.40 km²*

**Gauge TUROSZÓW**
km 1.0
A = 72.11 km²

*MIEDZIANKA RIVER*

**Gauge SIENIAWKA**
km 194.2
A = 693 km²

Fig.12. *Colors used to depict the two hydrographs are too similar and generate confusion*

We made them more distinguishable, as below.

[Figure]

---

## Referee Comment (RC2) · Anonymous Referee #2 · 23 Mar 2021

The discussed problem is highly interesting from a scientific perspective but principally in terms of operational safety of this type of structures. All aspects listed below require detailed explanations. If the above requirement is met, necessary information in ICOLD can be supplemented. There are numerous doubts at present. The study is a monograph and addresses a wide range of problems; this impedes arriving at conclusions for ICOLD and for an international Journal.

The following details form a set of key information necessary to analyze disasters of hydraulic structure, especially dams: 1. Functions to be performed by the structure – a description 2. Geomorphological and hydrological conditions 3. Design guidelines (applicable during design work), data adopted for designing purposes, obtained final flow capacity parameters of the structure, geotechnical parameters of the structure, device output curves 4. A short operational description of the structure, technical assessments made, hydrological events, structure condition (maintenance status), changes in geotechnical parameters, dislocation of land-surveying points, filtration through the structure and results of control operations 5. complete probabilistic and physical characteristics of the input function that directly caused the disaster 6. indirect conditions, here e.g. instructions for water management in the reservoir as a principal document binding upon the operator and deviations in control processes with their reasons 7. An analysis of simulation results and an assessment of potential differences compared to ICOLD data, applicable assessment methods that were used (e.g. empirical formulae) 8. If a structure with the same cross-section is to be reconstructed, a rationale must be given with applicable regulations and new characteristics of devices

Considering the number of problems addressed in the study, detailed comments can be compiled by April 2, 2021

---

## Author Comment (AC2) · 28 Apr 2021

**Response to Reviewers #2**

*RC: The discussed problem is highly interesting from a scientific perspective but principally in terms of operational safety of this type of structures. All aspects listed below require detailed explanations. If the above requirement is met, necessary information in ICOLD can be supplemented. There are numerous doubts at present. The study is a monograph and addresses a wide range of problems; this impedes arriving at conclusions for ICOLD and for an international Journal.*

Thank you very much for this general opinion. Following the suggestions of the reviewer we summarised additional data on the Niedów dam and on the conditioning of the catastrophe next to information already provided. We would also like to note that our paper has not been meant to be a full report of the Niedów dam failure or as a monograph (we consider this as a valuable suggestion). It is difficult to provide a full set of data relevant from different perspectives and interests in such a short communication. In the following, there are responses to the issues raised by the reviewer.

*1. Functions to be performed by the structure – a description*

The reservoir function is given, in lines 107-110:
The Niedów dam on the Witka river (in km 2.2) is located in the south-west Poland, near the Polish-Czech and Polish-German borders. It was constructed in 1962 to supply water to the Turów coal power station for cooling purposes and for drinking water supply to nearby settlements, including the town of Bogatynia. In essence, the reservoir function was not to mitigate the flood hazard.

*2. Geomorphological and hydrological conditions*

Basic information are given in the text (l. 112) and in Fig. 6. As the failure of the dam was due to the overtopping, details on geomorphological conditions are considered not that essential as it would be the case of other dam break causes.

It is further not clear what kind of additional information are meant by the reviewer in terms of hydrological conditions. They are described in section 2.2. and 2.3. The inflow hydrograph is further depicted in Fig. 13.

*3. Design guidelines (applicable during design work), data adopted for designing purposes, obtained final flow capacity parameters of the structure, geotechnical parameters of the structure, device output curves*

The former text l. 164-165 is extended into:

"Three tainter steel gates, with a width of 6.7 m and a height of 6.6 m each, controlled the water outflow from the reservoir, see Fig. 4. The maximum yield of the weir, when the gates are elevated by 5 m and water level in the reservoir reaches 210 m a.s.l., is 500,0 $m^3 s^{-1}$. This corresponds to the design flow with an exceedance probability of 1 %. This yield can reach a value of 655 $m^3 s^{-1}$ for the designed maximum water level of 210.4 m a.s.l. In addition, the pillars of the central section contained bottom outlets with size of 2m x 1m, equipped with vertically moving flat closures. The yield capacity of each outlet was 10 $m^3 s^{-1}$ at water level upstream of 210 m a.s.l. and 202,20 m a.s.l. downstream. In normal conditions these openings were utilized to empty the reservoir.

*4. A short operational description of the structure, technical assessments made, hydrological events, structure condition (maintenance status), changes in geotechnical parameters, dislocation of land-surveying points, filtration through the structure and results of control operations*

Relevant information is added to the text:

The dam was technically supervised regularly, and was stable and in good condition. A number of maintenance and restoration works were executed in the years from 1998 to 2009, including the repair of the steel and concrete structures, the repair of the upstream slope, and the replacing of the road pavement on the top of the dam in 2009.

*5. Complete probabilistic and physical characteristics of the input function that directly caused the disaster*

This comment is rather general and we can hardly meet it in our response and in the paper. We suppose that this may form a separate study, which is not the basic goal of the paper. Nevertheless, the flood magnitude is described by a reoccurrence period of 100-200 year, l. 129. The inflow hydrograph is further depicted in Fig. 13.

*6. Indirect conditions, here e.g. instructions for water management in the reservoir as a principal document binding upon the operator and deviations in control processes with their reasons*

We added information on the dam operational instructions:

The dam was operated accordingly to a complete dam documentation. In total there were five major documents: i) guidelines for the operation of the water intake, ii) guidelines for flood management in the reservoir area, iii) technical instruction of the dam operation during the flood, iv) manual for gate control, v) manual for the power plant operation.

During the catastrophic flood, to control the water level in the reservoir the crew initially followed the operational manual. The procedure was to gradually elevate the gate by 0,2 m in order to maintain the desired water level. When the control of one gate was insufficient, the additional gate was also raised by 0.2 m. In the course of this unpreceded water level rise, the gate opening was accelerated. However, the water level exceeded the edge of the repaired gate at the inlet to the hydroelectric power plant (which was undergoing renovation at the time). As a result, the control room was flooded, the crew was evacuated from the rooms to the top of the dam, and the power supply to the dam was turned off. Finally, the crew tried to open more gates manually from the dam's crest. The event took place on Sunday, which influenced the transmission of information. The crew did not have full knowledge of the scale of the flood and the damage in the territory of Chechia (incl. the information concerning the destruction of the Frydland water gauge station on the Smeda/Witka River above the reservoir).

*7. An analysis of simulation results and an assessment of potential differences compared to ICOLD data, applicable assessment methods that were used (e.g. empirical formulae)*

The authors performed an analysis and assessment on the dam breach dynamics in reference to several formulas available in the literature along with making use of hydrodynamic modelling. What kind of ICOLD data was meant by the reviewer is not clear. We do not feel obliged to obey the ICOLD

methodology, publicly not available. If desirable, it can be executed in other way, we will accept further kind suggestions.

*8. If a structure with the same cross-section is to be reconstructed, a rationale must be given with applicable regulations and new characteristics of devices*

The dam has been reconstructed by 2016. The new dam is a concrete dam equipped with an labyrinth overflow structure in place of the right side dam, as shown in figure below. Information on the design and construction of the new dam are in a technical report  Kostecki, S., Rędowicz, W. (2011). Physical model testing for the reconstruction of the Niedów dam (in Polish). Institute of Geo- and Hydraulic Engineering, Wrocław University of Science and Technology.

[Figure]

Figure 1: The Niedów dam after reconstruction (fot. Wojciech Rędowicz)